# Disparities in prostate cancer screening practices among general practitioners and urologists (PROSHADE study): A cross-sectional study

Blanca Lumbreras[1,2]*, Lucy A. Parker[1,2], Pablo Alonso-Coello[2,3], Juan-Pablo Caballero-Romeu[4,5], Ignacio Párraga-Martínez[6,7], Luis Prieto[4], Irene Moral-Peláez[8], Mª del Campo-Giménez[9], Luis Gómez-Pérez[10], Ana Cebrián[11], Maite López-Garrigós[2,12], Elena Ronda[2,13], Mercedes Guilabert[14], Carlos Canelo-Aybar[2,3], Ildefonso Hernández-Aguado[1,2]

1 Department of Public Health, History of Science and Gynecology, Miguel Hernandez University, San Juan de Alicante, Spain, 2 CIBER of Epidemiology and Public Health, CIBERESP, Madrid, Spain, 3 Iberoamerican Cochrane Centre - Department of Clinical Epidemiology and Public Health, Biomedical Re-search Institute Sant Pau, Barcelona, Spain, 4 Department of Urology, Dr. Balmis General University Hospital, Alicante, Spain, 5 Alicante Institute for Health and Biomedical Research (ISABIAL), Alicante, Spain, 6 Medical Sciences Department, Medicine Faculty, University of Castilla-La Mancha, Albacete, Spain, 7 Primary Care Research Group, Health Research Institute of Castilla-La Mancha (IDISCAM); Health Care Center Zone VIII, Servicio de Salud Castilla-La Mancha, Albacete, Spain, 8 EAP Sardenya, Barcelona, Institut de Recerca Sant Pau, Barcelona, Spain, 9 Integrated Care Management of Albacete, Health Service of Castilla-La Mancha, Barcelona, Spain, 10 Department of Urology, University General Hospital of Elche, Elche, Spain, 11 Cartagena Casco Healthcare Centre, Cartagena, Spain, 12 Clinical Laboratory, Hospital Universitario de San Juan, San Juan de Alicante, Spain, 13 Public Health Research Group, Alicante University, San Vicente del Raspeig, Spain, 14 Department of Health Psychology, Miguel Hernandez University, Elche, Spain

* blumbreras@umh.es

## Abstract

### Introduction

Recent European recommendations promote risk-based, patient-centred screening models that emphasise shared decision-making (SDM) in prostate cancer (PCa) screening with prostate-specific antigen (PSA) testing. Understanding how clinicians apply these practices is essential, particularly given the roles of general practitioners (GPs) and urologists in detection. The aim of this study was to compare the knowledge, attitudes, and practices of GPs and urologists in Spain regarding PSA testing, PCa screening, and SDM implementation.

### Methods

Cross-sectional survey conducted via an online questionnaire. Members of the Spanish Association of Urology (AEU) and the Spanish Society of Family and Community Medicine (semFYC) were surveyed online. The survey, developed and validated using a modified Delphi process, contained 18 items assessing clinicians' opinions,

**Data availability statement:** The dataset generated and analyzed during the current study is available in Zenodo at https://doi.org/10.5281/zenodo.19983225.

**Funding:** Research was funded by the Instituto de Salud Carlos III, code PI20/01334, co-financed with FEDER funds from the European Union "A way of doing Europe". The funders had no role in study design, data collection and analysis, decision to publish, or preparation of the manuscript.

**Competing interests:** No financial disclosures were reported by the authors of this paper.

practices, and knowledge of guidelines. Data from 494 respondents (280 GPs and 214 urologists) were analysed using descriptive statistics and logistic regression to explore differences between specialty, demographics, and screening practices.

## Results

Urologists were more likely than GPs to consider PSA testing important (97.2% vs. 83.2%) and useful (96% vs. 44.4%), and to recommend it to relatives (90.4% vs. 44%) (all $p < 0.001$). They reported to initiate testing earlier (50–59 years: 77.8% vs. 32%) and reported to perform it more frequently (annual testing: 60.7% vs. 45.6%, $p < 0.001$). GPs reported to more often engaged in SDM when ordering PSA tests (69.7% vs. 10%, $p < 0.001$). Knowledge of guidelines was higher among urologists (91% vs. 24.1%), yet 75.4% stated that they had not modified practice following updated recommendations.

## Conclusions

Significant specialty-based differences exist in PSA testing practices, perceptions, and guideline adherence. Urologists show greater familiarity with recommendations, while GPs reported more patient-centred communication.

## Introduction

Cancer remains one of the leading causes of mortality in Europe, and poses major health, social, and economic challenges. The European Commission launched Europe's Beating Cancer Plan (EBCP) in 2021, an ambitious, patient-centred strategy aimed at reducing the cancer burden through coordinated actions in prevention, early detection, diagnosis and treatment [1]. One of the priorities of this plan is to promote cancer screening through risk-based and personalised approaches.

Prostate cancer (PCa), the most frequently diagnosed malignancy among men in Europe, is one of the main targets of this strategy. Although the prostate-specific antigen (PSA) test has contributed to earlier diagnosis, its limited specificity has also led to substantial overdiagnosis and overtreatment [2]. To address these challenges, the European Association of Urology (EAU) has proposed a multi-risk algorithm that integrates PSA testing within a broader risk-adapted model for early detection. This approach combines PSA results with individual factors such as age, family history, ethnicity, genetic predisposition, and, when appropriate, complementary assessments including multiparametric magnetic resonance imaging (MRI) and molecular biomarkers [3]. Recent policy developments, such as the EUCanScreen programme [4], reflect a coordinated effort to improve the quality, accessibility, and effectiveness of cancer screening programmes across Member States. In parallel, the Council Recommendation of 9 December 2022 [5] on strengthening prevention through early detection provides updated guidance that addresses the need to expand and refine screening strategies beyond certain specific cancers. This recommendation identifies

PCa screening as a key area for further development, acknowledging both the preliminary evidence and the widespread use of opportunistic screening practices across Europe. It encourages countries to adopt a gradual and evidence-based approach, through the implementation of pilot programs with the generation of additional research. Furthermore, it underlines the importance of evaluating PSA testing in combination with MRI as a follow-up diagnostic tool, particularly within the context of organised screening programmes, in order to assess feasibility, effectiveness, and potential benefits versus harms.

This represents a sensible and patient-centred strategy that balances the benefits of early detection with the potential harms of overdiagnosis and overtreatment. Thus, the implementation of shared decision-making (SDM) in this strategy is crucial. SDM ensures that men are fully informed about the benefits, limitations, and potential outcomes of PCa screening, which empowers them to participate actively in decision-making. However, its implementation in clinical practice varies widely [6,7].

Both general practitioners (GPs) and urologists play critical roles in implementing this new approach. GPs are typically the first contact point for men considering screening and are responsible for initiating informed discussions [8], while urologists provide specialist evaluation, diagnostic interpretation, and management of abnormal findings [9]. These incidental findings also require an approach that includes clear communication with patients and timely referral to appropriate specialities (e.g., radiology, oncology or other relevant disciplines), to ensure personalised assessment and follow-up which minimize unnecessary interventions and patient anxiety [10].

Effective collaboration and alignment between these two professional groups are essential to ensure consistency, accuracy, and patient trust in clinical recommendations [11]. In addition, urologists have access to a wider range of diagnostic options than GPs, including the use of additional biomarkers and multi-parametric MRI, and are therefore, more likely to integrate these risk-adapted approaches [12]. However, GPs usually base themselves solely on PSA levels, leading to variability in clinical decisions. A study on PSA testing practices found that GPs were less likely to routinely order PSA tests compared to urologists, largely due to concerns regarding false positive results, overdiagnosis, and the resulting potential psychological burden on patients [11]. In addition, although this manuscript focuses primarily on the roles of GPs and urologists, it is important to recognize the broader governance framework needed for effective implementation of organized screening programs. In particular, public health specialists play a central role in ensuring adherence to protocols, population coverage, and the promotion of informed participation [13]. In addition, other clinical specialists such as radiologists and pathologists are essential for accurate imaging interpretation, diagnostic confirmation, and risk stratification in the screening pathway [14]. Moreover, the involvement of professional associations and patient organizations, in order to align patient needs, is essential for the cocreation of screening strategies [15].

In this context, assessing the knowledge, attitudes, and practices of GPs and urologists regarding PSA use, screening, and SDM becomes a necessary step for implementing the EAU and EBCP recommendations in routine clinical practice. Conducting a survey among GPs and urologists can provide valuable information about their familiarity with screening practices, PSA use, and their communication strategies for addressing the benefits and risks of screening. It can also identify differences in understanding, perceived barriers, and training needs that may influence clinical practice. Although several previous surveys have examined clinicians' knowledge and practices regarding PCa screening [16] it remains necessary to reassess this knowledge in the context of the most recent European recommendations.

Therefore, the present study aims to explore and compare the knowledge, attitudes, and practices of GPs and urologists regarding PSA testing, PCa screening and SDM. By identifying common barriers and differences between both groups, the study seeks to provide evidence to guide educational strategies and facilitate the successful implementation of the European risk-based screening model in clinical practice.

## Materials and methods

The protocol of the PROSHADE study has been previously published [17]. The reporting of the manuscript follows the STROBE [18] recommendation (S1 Table).

## Design

Cross-sectional survey based on an online questionnaire.

## Questionnaire

The survey was developed following a systematic review of the literature [16] Based on this, the coordinating group drafted an initial version of the questionnaire, which was refined and validated using a modified Delphi method. A panel of experts—including GPs, epidemiologists, urologists, a lab physician, and a psychologist—reviewed the questionnaire in two rounds of virtual voting, rating item relevance and suggesting improvements. In the final phase, a pilot study was conducted with the expert panel and 10 family doctors and urologists. No major changes were needed after the pilot. The questionnaire validation process included assessing the degree of consensus regarding the relevance of item for inclusion in the final questionnaire. Content validity was analyzed through item averages, Aiken's V test, and qualitative expert assessments to adjust questionnaire categories. Criteria for item selection were a mean score above 3.5 and Aiken's V test result greater than or equal to 0.70. Items with a lower 95% confidence interval limit below 0.70 were included if the mean exceeded 3.5 and the median was 4 or higher. We collected the expert's evaluations using Google Forms and analyzed the data with IBM SPSS Statistics v27 for Windows. The final version of the questionnaire included 18 questions related to opportunistic PCa screening and the PSA testing: 4 questions about clinician's opinions, 10 questions about routine practice and 4 questions about knowledge and use of current guidelines. Answers were all presented as Likert scale (Supplementary S2).

## Subjects and procedure

Clinicians (urologists and GPs) were selected through collaboration with two major Spanish scientific societies: the Spanish Association of Urology (AEU) and the Spanish Society of Family and Community Medicine (semFYC). These societies were contacted and agreed to support the study by facilitating access to their members. An invitation to participate was distributed to urologists and GPs through internal mailing lists and communication channels managed by each society. The invitation included a brief description of the study objectives, participation requirements, and a link to the online survey. The survey was administered using a Google Forms questionnaire, allowing for distribution, anonymous responses, and centralized data collection. Survey was carried out between: 01/05/2024–31/07/2024. We did not have access to information that could identify individual participants during or after data collection.

## Sample size

The aim of the study was to assess the knowledge, opinions, and implementation of SDM in practice of GPs and urologists regarding PCa and PSA testing in opportunistic PCa screening. We focused on the implementation of SDM in practice as the main outcome for calculating the sample size (since it is the most unfavorable outcome of the three proposed objectives). According to a recent systematic review [16], a minimum of 384 participants would be required to estimate this proportion with a precision of 5% at a 95% confidence level under a simple random sampling assumption. However, as participants were recruited through a convenience sampling approach—consisting of clinicians who voluntarily responded to the survey invitation distributed via scientific societies (there are 1,630 urologists included in the AEU and 20,000 GPs in the semFYC)—this calculation was used only as a reference to ensure an adequate sample size rather than to support formal precision estimates. Recruitment was therefore continued until a sufficient number of responses from both specialties was obtained to allow meaningful comparisons between groups.

## Variables included

The independent variables included in the questionnaire were: sex, age (years), specialty (GPs and urologists), years of practice (including specialty), regular formation received regarding the PSA test (defined as having received prior formal

or structured training on PSA testing, including its indications, interpretation, and clinical management, through continuing medical education activities such as courses, workshops, or similar programs), and type of centre where the clinician works (public, private or both).

## Statistical analysis

The data collected in the study was coded and recorded in a specifically designed database. Analysis was performed using IBM SPSS Statistics 27.0 (IBM Corp., Armonk, NY, USA).

We assume that missing values occurred at random (there were no differences between those cases who answered a particular question and those who did not) and thus, we did not omit those cases with missing data and analyze the remaining data). However, to explore this assumption, we compared key characteristics (sex, specialty, and years of practice) between participants with complete data and those with missing responses for the main variables, and no relevant differences were observed. Sensitivity analyses were conducted using complete-case datasets, and the results were consistent with the primary analyses.

The descriptive statistics for categorical variables were expressed in numbers and percentages. The association between the independent variables included in the questionnaire (sex, age, specialty, years of practice) and the clinicians' opinions about opportunistic PCa screening and the PSA testing were tested using the Chi-square test. This test was also used for the analysis of differences in routine practice in PSA screening for prostate cancer by specialty and also for the analysis of the differences in the use of available guidelines according to the information provided on the advantages and disadvantages of PSA determination and specialty. A p-value <5% was considered statistically significant. Continuous variables were assessed for normality and, as they were not normally distributed, are presented as medians and interquartile ranges; comparisons between groups were performed using the Mann–Whitney U test.

Given the number of Chi-square tests performed, we acknowledge the potential for increased Type I error. These analyses were exploratory and aimed at identifying patterns between specialties rather than testing a single predefined hypothesis; therefore, unadjusted p-values were reported to facilitate interpretation. To further assess the robustness of the findings and account for potential confounding, we conducted multivariable logistic regression analyses for the main outcomes, including variables such as specialty, sex, years of practice, and prior PSA training. These models allowed us to estimate adjusted associations between predictors and outcomes. The results of these analyses were consistent with the univariate findings, particularly for SDM and guideline use. In addition, applying a Bonferroni correction ($\alpha = 0.05$ divided by approximately 25 comparisons; adjusted $\alpha \approx 0.002$) did not change the interpretation, as the main associations remained statistically significant.

All questionnaire items were originally measured using Likert-type scales. For descriptive and comparative analyses, some variables were grouped into broader categories to facilitate interpretation. In particular, selected variables were dichotomized into low versus high categories based on conceptual relevance. Sensitivity analyses using the original ordinal scales showed consistent results.

## Ethic statement

This study did not involve human subjects, identifiable personal data, or clinical interventions. Therefore, formal compliance with the Declaration of Helsinki was not required. Nevertheless, ethical principles for voluntary participation and data protection were followed throughout the study.

Participation in the Delphi survey was voluntary. Potential participants were invited via professional mailing lists and received information about the study objectives, procedures, and data handling. Completion of the anonymous online questionnaire (administered via Google Forms) was considered as informed consent. No personally identifiable information was collected at any stage. Institutional Review Board Statement: CEIC Sant Joan d'Alacant (20/041) on 8th January 2021.

## Results

The study included 494 clinicians, GPs (280, 56.7%) and urologists (214, 43.3%) (Table 1). The sample included 228 men (46.2%) and 257 women (52%), with a median age of 49 years (IQR: 35.8–58) and a median of 22 years of professional experience (IQR: 10–30). Most of the surveyed clinicians had received formal education regarding the PSA test (303, 61.3%). Most of the surveyed clinicians worked in the public sector (470, 95%): the urologists all worked in a hospital setting while the GPs worked in primary care centres.

### Evaluation of the differences in clinicians' opinions regarding opportunistic PCa screening and PSA testing according to relevant variables

Participants expressed differing opinions regarding opportunistic PCa screening and PSA testing, according to sociodemographic variables (Table 1).

Men stated that were more likely to recommend the PSA test to their relatives than women (66.9% vs. 54.4%, $p = 0.012$), although there were no significant differences by sex in terms of concern about missing PCa, perceived importance of PCa screening, or perceived usefulness of the PSA test. Participants with more years of practice expressed significantly greater concern about not detecting PCa (median 23 years vs. 16 years; $p = 0.016$). Those who had previously received formal education about the PSA test were significantly less concerned about missing PCa (73.1% vs. 87.8%, $p = 0.001$) and considered the PSA test less useful for diagnosis (63.4% vs. 48.7%, $p = 0.007$). Urologists were significantly more likely than general practitioners to rate PCa screening as important (97.2% vs. 83.2%, $p < 0.001$), to consider the PSA test useful (96% vs. 44.4%, $p < 0.001$), and to state that they would recommend it to their relatives (90.4% vs. 44%, $p < 0.001$).

In the multivariable analysis, after adjusting for previous training on PSA testing, years of practice remained significantly associated with greater concern about missing PCa (aOR = 1.023; 95% CI: 1.005–1.044; $p = 0.024$). In addition, after adjusting for previous training on PSA testing, urologists were significantly more likely than GPs to consider PSA testing useful (aOR = 0.037; 95% CI: 0.013–0.084; $p < 0.001$). However, after adjusting for sex, there were no significant differences between specialties in the willingness to recommend PSA testing to relatives.

### Analysis of differences in routine practice in PSA screening for PCa by specialist area

Differences in self-reported use of PSA testing were observed between urologists and GPs (Table 2). Urologists reported recommending PSA testing at earlier stages more frequently (40–49 years: 20, 12.3% and 50–59 years: 126, 77.8%) compared to GPs (40–49 years: 10, 3.6% and 50–59 years: 89, 32%), who more frequently reported not recommending the test at all in asymptomatic patients (160, 57.6% vs 173, 39.3%), $p < 0.001$.

Both groups reported requesting the highest number of PSA tests primarily for patients aged 60–69 years (112, 70% vs 152, 62%, respectively). However, GPs reported a greater tendency to request PSA tests for patients aged 70–79 years compared with urologists (18, 7.3% vs 2, 1.3%, respectively; $p < 0.001$).

When deciding the appropriate age to stop PSA testing, GPs reported a tendency to favour stopping earlier (49, 17.6% at 70 years) compared to urologists (13, 9.1%), $p < 0.001$.

Urologists reported greater consistency with annual PSA testing practices than GPs (82, 60.7% vs 124, 45.6%, respectively), ($p < 0.001$), and significantly more likely to report routinely offering PSA tests to patients undergoing blood tests for unrelated reasons in comparison with GPs (79, 61.7% vs. 74, 26.6%, respectively) ($p < 0.001$). In addition, urologists reported a higher likelihood of requesting a PSA test without prior explanation when actively requested by the patient compared with GPs (13, 9.8% vs. 8, 2.9%, respectively) ($p = 0.03$). Urologists reported lower engagement in SDM when ordering a PSA test compared with GPs (12, 10% vs. 193, 69.7%, respectively) ($p < 0.001$). In the multivariable analysis, adjusting for variables that were significant in the univariate analysis (sex and specialty), GPs were significantly more likely to engage in SDM than urologists (aOR = 5.345; 95% CI: 2.793–10.568; $p < 0.001$).

Table 1. Differences in clinicians' opinions about opportunistic PCa screening and the PSA testing according to sociodemographic variables and practice related variables.

| Variables (n, %) | Total | How concerned are you about not detecting PCa? | | | Importance of PCa screening? | | | Usefulness of the PSA test for PCa diagnosis | | | Would you recommend the PSA test to your relatives? | | |
|---|---|---|---|---|---|---|---|---|---|---|---|---|---|
| | | None or little | Quite a lot or a lot | p-value | None or little | Quite a lot or a lot | p-value | None or little | Quite a lot or a lot | p-value | None or little | Quite a lot or a lot | p-value |
| **Sex** | | | | 0.450 | | | 0.152 | | | 0.097 | | | 0.012 |
| Man | 228 | 44 (24.7) | 134 (75.3) | | 20 (8.8) | 208 (91.2) | | 57 (34.8) | 107 (65.2) | | 58 (33.1) | 117 (66.9) | |
| Woman | 257 | 50 (21.6) | 182 (78.4) | | 33 (12.8) | 224 (87.2) | | 99 (43) | 131 (57) | | 103 (45.6) | 123 (54.4) | |
| NA | 9 | | | | | | | | | | | | |
| **Age (years) (median, IQR)** | 49 (35.8-58) | 47 (33-56) | 50 (37-59) | 0.292 | 46 (32-57) | 50 (36-58) | 0.073 | 49 (37-57) | 48 (34-58) | 0.370 | 47 (36-57) | 50 (35-59) | 0.325 |
| **Years of practice (years) (median, IQR)** | 22 (10-30) | 16 (9-26) | 23 (10-30) | 0.016 | 19 (9-28) | 22 (10-30) | 0.638 | 24 (11-30) | 21 (9-30) | 0.336 | 22 (11-30) | 21 (9-30) | 0.935 |
| **Regular formation received regarding the PSA test** | | | | 0.001 | | | 0.213 | | | 0.007 | | | 0.350 |
| No | 119 | 14 (12.2) | 101 (87.8) | | 11 (9.2) | 108 (90.8) | | 60 (51.3) | 57 (48.7) | | 53 (45.7) | 63 (54.3) | |
| Yes | 307 | 77 (26.9) | 209 (73.1) | | 42 (13.7) | 265 (86.3) | | 98 (36.6) | 170 (63.4) | | 112 (40.6) | 164 (59.4) | |
| NA | 68 | | | | | | | | | | | | |
| **Speciality** | | | | 0.168 | | | <0.001 | | | <0.001 | | | n<0.001 |
| Urologist | 214 | 27 (18.6) | 118 (81.4) | | 6 (2.8) | 208 (97.2) | | 5 (4) | 120 (96) | | 13 (9.6) | 123 (90.4) | |
| General Practitioner | 280 | 67 (24.5) | 206 (75.5) | | 47 (16.8) | 233 (83.2) | | 154 (55.6) | 123 (44.4) | | 153 (56) | 120 (44) | |
| **Total** | **494** | **94 (25)** | **324 (75)** | | **53 (10.7)** | **441 (89.3)** | | **159 (39.6)** | **243 (60.4)** | | **166 (40.6)** | **243 (59.4)** | |

NA: Not available; IQR: Interquartile Range; PSA: Prostate Specific Antigen; PCa: Prostate cancer.

**Table 2. Analysis of differences in routine practice in PSA screening for prostate cancer by speciality.**

| Questions | Total (494) (n, %) | General Practitioners (280) (n, %) | Urologists (214) (n, %) | p-value |
|---|---|---|---|---|
| **At what age would you request the first PSA test in an asymptomatic man?** | | | | *<0.001* |
| 40-49 | 30 (6.8) | 10 (3.6) | 20 (12.3) | |
| 50-59 | 215 (48.9) | 89 (32.0) | 126 (77.8) | |
| Over 60 | 22 (5.0) | 19 (6.8) | 3 (1.9) | |
| Would not request | 173 (39.3) | 160 (57.6) | 13 (8.0) | |
| NA | 54 | | | |
| **In which age decade do you apply for the highest number of PSA requests?** | | | | *0.004* |
| 40-49 | 2 (0.5) | 0 (0.0) | 2 (1.3) | |
| 50-59 | 119 (29.4) | 75 (30.6) | 44 (27.5) | |
| 60-69 | 264 (65.2) | 152 (62.0) | 112 (70.0) | |
| 70-79 | 20 (4.9) | 18 (7.3) | 2 (1.3) | |
| NA | 89 | | | |
| **In which age decade do you think PSA is most sensitive for diagnosing prostate cancer?** | | | | *<0.001* |
| 40-49 | 24 (5.7) | 17 (6.1) | 7 (4.9) | |
| 50-59 | 135 (32.2) | 76 (27.4) | 59 (41.5) | |
| 60-69 | 157 (37.5) | 95 (34.3) | 62 (43.7) | |
| 70-79 | 28 (6.7) | 17 (6.1) | 11 (7.7) | |
| NS/NC | 75 (17.9) | 72 (26.0) | 3 (2.1) | |
| NA | 75 | | | |
| **At what age do you think you should stop ordering PSA in a male with normal PSA values?** | | | | *<0.001* |
| 70 | 62 (14.7) | 49 (17.6) | 13 (9.1) | |
| 75 | 112 (26.5) | 58 (20.8) | 54 (37.8) | |
| 80 | 148 (35.1) | 89 (31.9) | 59 (41.3) | |
| Never | 32 (7.6) | 23 (8.2) | 9 (6.3) | |
| None of the above | 68 (16.1) | 60 (21.5) | 8 (5.6) | |
| NA | 72 | | | |
| **How often would you order a PSA test for a male with a previous apparently normal level, and within the appropriate age range?** | | | | *<0.001* |
| Every year or less | 206 (50.6) | 124 (45.6) | 82 (60.7) | |
| Every two years | 127 (31.2) | 84 (30.9) | 43 (31.9) | |
| Every two years or more | 74 (18.2) | 64 (23.5) | 10 (7.4) | |
| NA | 87 | | | |
| **If a patient actively requests the PSA test, I...** | | | | *0.03* |
| Request it without explanation. | 21 (5.1) | 8 (2.9) | 13 (9.8) | |
| I inform about the (dis)advantages of the test and, if indicated, request it. | 259 (63.0) | 171 (61.3) | 88 (66.7) | |
| I inform about the (dis)advantages of the test and request it, even if it is not indicated. | 117 (28.5) | 88 (31.5) | 29 (22.0) | |
| Other | 14 (3.4) | 12 (4.3) | 2 (1.5) | |
| NA | 83 | | | |
| **How many PSA tests per year would you order in an asymptomatic 65-year-old male with no treatment and a last test of 3 ng/ml performed one year ago?** | | | | *<0.001* |
| None | 116 (28.3) | 104 (37.4) | 12 (9.1) | |
| One | 236 (57.6) | 144 (51.8) | 92 (69.7) | |
| Two | 54 (13.2) | 29 (10.4) | 25 (18.9) | |

*(Continued)*

**Table 2.**  (Continued)

| Questions | Total (494) (n, %) | General Practitioners (280) (n, %) | Urologists (214) (n, %) | p-value |
|---|---|---|---|---|
| Three | 4 (1.0) | 1 (0.4) | 3 (2.3) | |
| NA | 84 | | | |
| **Do you offer PSA testing to men who come for another reason and are going to have a blood test?** | | | | *<0.001* |
| No | 253 (62.3) | 204 (73.4) | 49 (38.3) | |
| Yes | 153 (37.7) | 74 (26.6) | 79 (61.7) | |
| NA | 88 | | | |
| **If you have decided that PSA is indicated in a particular patient:** | | | | *0.423* |
| I include it in the test without asking the patient. | 108 (26.7) | 75 (27.1) | 33 (26.0) | |
| I ask the patient if he/she wants the test and explain its advantages and disadvantages. | 279 (69.1) | 188 (67.9) | 91 (71.7) | |
| I give the patient a standardised written form/ Other | 17 (4.2) | 14 (5.1) | 3 (2.4) | |
| NA | 90 | | | |
| **Is the decision to order a PSA test shared with the patient?** | | | | *<0.001* |
| Never/almost never | 91 (22.9) | 20 (7.2) | 71 (59.2) | |
| Sometimes | 101 (25.4) | 64 (23.1) | 37 (30.8) | |
| Often/always | 205 (51.6) | 193 (69.7) | 12 (10.0) | |
| NA | 97 | | | |

NA: Not available; PSA: Prostate Specific Antigen.

### Differences in the information provided by the clinicians to the patients regarding the advantages and disadvantages of the PSA test

Clinicians' self-reported knowledge and adherence to relevant guidelines varied according to specialist area (Table 3). Urologists reported greater knowledge of at least one of the following guidelines: EAU/AEU or (Preventive Activities and Health Promotion Program of the Spanish Society of Family and Community Medicine (semFYC) (PAPPS) guidelines [19] than GPs, and also a greater knowledge of the USPSTF guidelines [5] (p<0.001 in both comparisons).

Urologists also reported a higher likelihood of applying these recommendations from these organizations in their daily practice (92.7%) compared to GPs (36.2%). Despite their familiarity with the guidelines, a considerable majority of clinicians reported not having changed their PSA testing practices following the update of the EAU guidelines in 2021 (75.4% had not changed), although urologists showed a higher rate of update (34.9%) compared to GPs (19.5%).

Fig 1 presents a comparative analysis of how healthcare professionals from different specialist areas, and according to their awareness of existing clinical guidelines, report informing patients about the advantages of PSA testing. This figure shows that GPs who reported being aware of the guidelines were more likely to report informing patients with information about the benefits of PSA testing, such as early detection of PCa, than those who were not. Fig 2 focus on the disadvantages. It shows that GPs who reported awareness of the guidelines were more likely to report discussing potential disadvantages – such as overdiagnosis, false positives or unnecessary treatments – with the patients, compared to those who were not. In the multivariable analysis, adjusting for variables that were significant in the univariate analysis (sex, specialty, and previous training on the PSA test), urologists were significantly more likely to follow the guidelines than GPs (aOR = 0.046; 95% CI: 0.022–0.088; p<0.001). In addition, previous training on PSA testing was associated with higher adherence to guidelines (aOR = 2.011; 95% CI: 1.109–3.750; p=0.024).

 

**Table 3. Knowledge and use of current guidelines on early detection tests for prostate cancer.**

| Questions | Total = 494 (n, %) | General Practitioner = 280 (n, %) | Urologist (214) (n, %) | p value |
|---|---|---|---|---|
| **Knowledge of the content of the following guides:** | | | | |
| **European guideline on prostate cancer (European Association of Urology), Spanish Association of Urology or Preventive Activities and Health Promotion Program of the Spanish Society of Family and Community Medicine** | | | | <0.001 |
| Yes, they know the content | 206 (48.8) | 64 (24.1) | 142 (91) | |
| Yes, they have read the guide, but they do not know exactly the content | 59 (14) | 48 (18) | 11 (7.1) | |
| Yes, they have heard of it | 46 (10.9) | 45 (16.9) | 1 (0.6) | |
| No, they don't know the content | 111 (26.3) | 109 (41) | 2 (1.3) | |
| NA | 73 | | | |
| **US Preventive Services Task Force (USPSTF)** | | | | <0.001 |
| Yes, they know the content | 130 (24.5) | 44 (16.2) | 59 (39.6) | |
| Yes, they have read the guide, but they do not know exactly the content | 80 (19.0) | 66 (24.4) | 14 (9.4) | |
| Yes, they have heard of it | 83 (19.8) | 67 (24.7) | 16 (10.7) | |
| No, they don't know the content | 154 (36.7) | 94 (34.7) | 60 (40.3) | |
| NA | 74 | | | |
| **Use of the available recommendations in usual practice.** | | | | <0.001 |
| Yes | 236 (56.5) | 97 (36.2) | 139 (92.7) | |
| No | 182 (43.5) | 171 (63.8) | 11 (7.3) | |
| NA | 76 | | | |
| **Change in the way they use the PSA test since the publication of the latest recommendations from the European Association of Urology** | | | | 0.001 |
| Yes | 96 (24.6) | 51 (19.5) | 45 (34.9) | |
| No | 295 (75.4) | 211 (80.5) | 84 (65.1) | |
| NA | 103 | | | |

NA: Not available; PSA: Prostate Specific Antigen.

## Discussion

The present study found significant differences between how GPs and urologists reported to deal with PCa opportunistic screening in Spain. In general, urologists expressed more favorable opinions toward the importance, usefulness, and recommendation of PSA testing compared to GPs. Urologists were also more inclined to initiate PSA testing in patients of younger ages, asymptomatic patients, and to test those in older age groups. Additionally, urologists reported ordering more frequent routine annual PSA tests and to offer these tests when patients had unrelated blood tests. GPs reported a higher level of engagement in SDM processes when ordering PSA tests compared to urologists. Urologists also reported a considerably higher level of knowledge regarding existing clinical guidelines, compared to GPs, and reported applying them in their daily practice more frequently. It is important to point out that knowledge of clinical guidelines was related to a greater likelihood of providing more complete and balance of information communicated to patients, regarding PSA testing. However, GPs more frequently reported to implement SDM when ordering a PSA test. This apparent paradox may be explained by several factors. First, urologists often work in more specialised clinical settings, where care is more focused on confirming diagnoses and making management decisions, which may limit opportunities for SDM [20]. Second, because they are more familiar with PCa and its consequences, they may feel more confident about the benefits of testing and therefore be less likely to actively involve patients in the decision-making process. Third, differences in professional

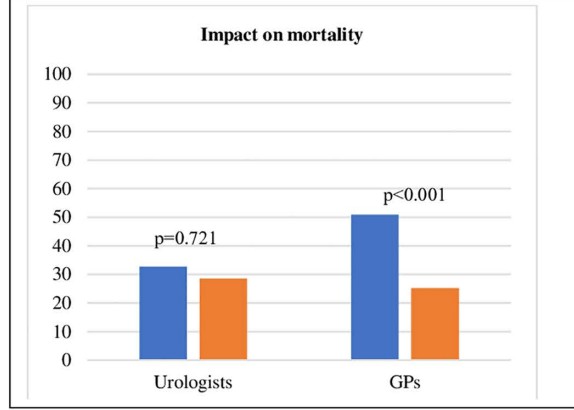
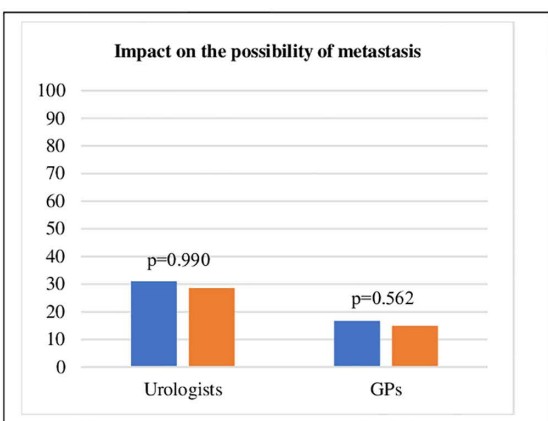
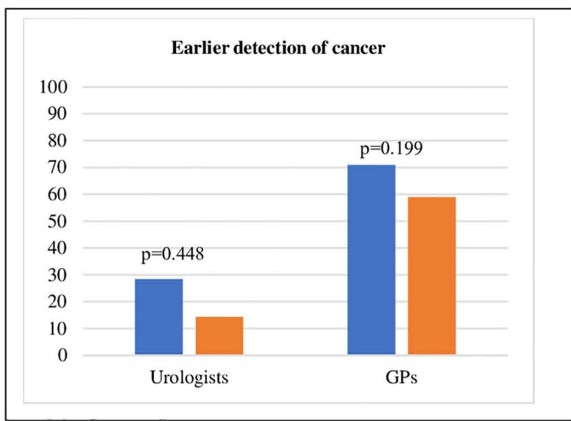
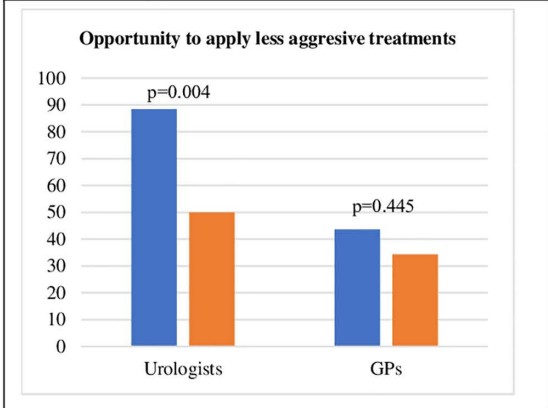

**Fig 1. Analysis of the differences in the provision of information provided regarding the advantages of the PSA test according to the reported knowledge of available guidelines by each speciality.**

culture may also contribute, as primary care tends to place greater emphasis on holistic care and patient-centred communication, including discussion of risks and patient preferences [21]. Importantly, this does not reflect a lack of competence, but rather highlights an opportunity to better support SDM in specialised settings, for example through improved communication tools and targeted training.

The data from our study highlights significant variability among clinicians' self-reported practices regarding opportunistic PCa screening and attitudes towards PSA testing, influenced by sociodemographic factors and clinical specialty. Similar trends have been observed in existing literature, where clinicians' opinions vary widely based on training, information received, and professional role [22,23]. In our study, clinicians with more years of practice showed greater concern about missing PCa, consistent with findings suggesting that experienced clinicians may rely more on routine screening to mitigate the risks of missed diagnoses [24]. In addition, clinicians who reported receiving formal education on PSA testing were less likely to express concern about underdiagnosis of PCa, which should not be interpreted as a greater inclination toward screening but rather as reflecting a more cautious and balanced perspective. This finding, together with their lower perceived diagnostic utility of the test and greater attention to communicating its limitations, suggests that education may promote a more critical appraisal of both the benefits and harms of PSA screening. In this context, while GPs appear to more consistently incorporate the communication of risks and uncertainties into clinical practice, urologists—who often

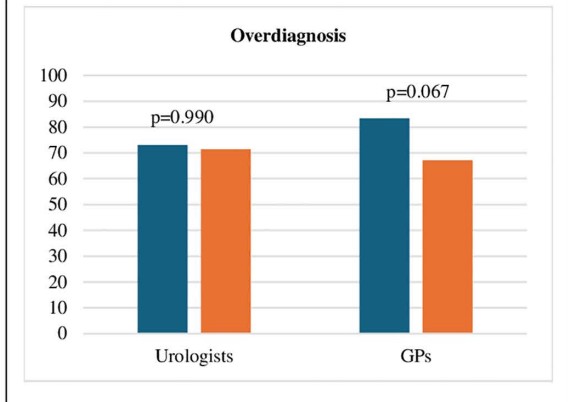
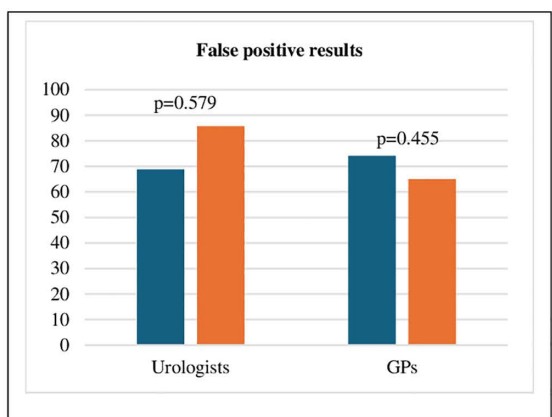
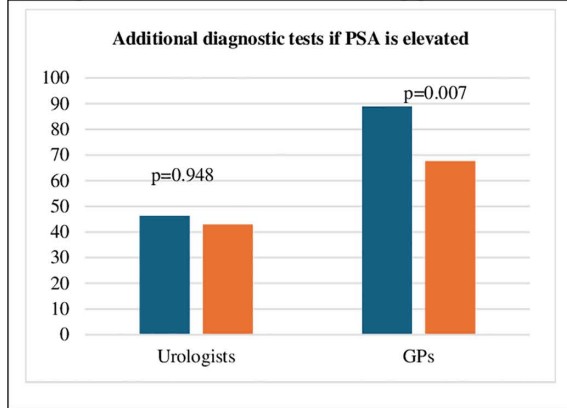
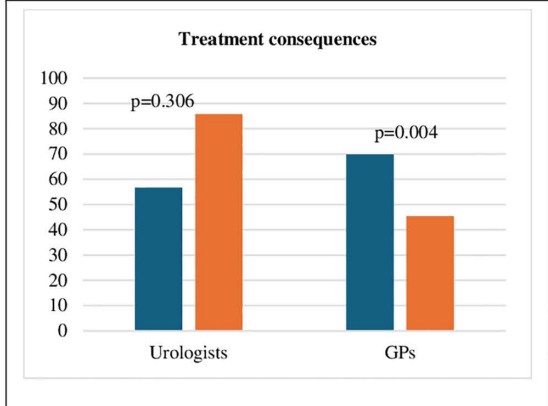

**Fig 2. Analysis of the differences in the provision of information provided regarding the disadvantages of the PSA test according to the reported knowledge of available guidelines by each speciality.**

work in more specialised settings—may benefit from additional support and tools to enhance patient communication, particularly regarding issues such as overdiagnosis and overtreatment [25].

Specialization substantially influenced perspectives on PSA testing, with urologists reporting a greater inclination to support the utility of PSA screening and to recommend it to relatives compared to GPs. This aligns with previous research suggesting that specialists are often advocates of routine PSA screening due to their tendency to minimize the potential risks of screening, as reflected in their differing values compared with GPs. In contrast, GPs may adopt more conservative approaches due to concerns over potential overtreatment and overdiagnosis [26,27].

Differences in self-reported routine clinical practices were also observed between urologists and GPs. Urologists reported a tendency to initiate PSA screening at younger ages, to request PSA tests more frequently, and to extend screening into older age groups compared to GPs. These patterns may be influenced by differences in guideline recommendations, such as those from the European Association of Urology guidelines, which recommend early and frequent screening practices for certain high-risk populations, in contrast to the more restrictive approach recommended by the US Preventive Services Task Force [7] or the Preventive Activities and Health Promotion Program (PAPPS) of the Spanish Society of Family and Community Medicine (semFYC) [19]. In this context, it is also important to acknowledge that different types of guideline-producing bodies may adopt distinct methodological approaches; for example, public health-oriented organisations such as the US Preventive Services Task Force aim to minimise potential conflicts of interest by

limiting the direct involvement of specialists in decision-making panels [28], whereas specialty-driven guidelines may reflect a more focused clinical perspective. This represents yet another case of how differences in values and preferences can lead to divergent clinical decisions — the same evidence interpreted through different professional perspectives results in different approaches to patient care.

Self-reported knowledge and use of clinical guidelines varied between specialties. Urologists reported significantly greater knowledge of and adherence to the guidelines than GPs. GPs reported lower familiarity with the European Association of Urology guidelines, which reflect the recent recommendations for PCa screening, and the US Task Force recommendations are distant and rarely consulted in daily practice. However, clinicians who reported being familiar with the guidelines were more likely to report informing patients of the advantages and disadvantages of PSA screening, supporting previous findings that guideline awareness could improve SDM [29].

The disparities identified in the study between GPs and urologists have important implications for clinical practice, patient outcomes and health policy. Differences in self-reported attitudes, practices and knowledge of guidelines suggest the need for specific interventions aimed at improving dissemination of guidelines, clarifying recommendations and promoting consistent shared decision-making processes in both primary care and specialised urological settings. Educational programmes specifically designed to address GPs' concerns about PSA screening and equip them with tools for effective communication with patients could help overcome these gaps. Furthermore, harmonisation of guideline recommendations across medical organisations and clear communication of their practical implications would mitigate perceived complexity and conflicts, potentially improving adherence and consistency in clinical practice. Thus, our findings underscore the importance of ongoing education and guideline dissemination to reduce practice variability and promote informed dialogues between patients and clinicians. Continued efforts to standardise PSA screening practices through professional education would also increase consistency and potentially improve patient outcomes.

The research is not without limitations. The cross-sectional design limits the ability to accurately capture practice patterns over time. In particular, while associations were observed between factors such as guideline knowledge or formal education and reported clinical behaviours, the study design does not allow us to determine whether such knowledge leads to changes in practice. Therefore, these findings should be interpreted as correlations rather than evidence of causation, and longitudinal studies would be needed to assess the direction and impact of these relationships. Reliance on self-reported data could introduce biases, potentially leading to overestimation or underestimation of actual clinical practices. Recruiting participants exclusively through scientific societies may introduce volunteer bias, as affiliated clinicians may be more engaged with continuing education, guideline updates, and professional development than non-affiliated practitioners. However, this approach provided access to a large and geographically diverse sample: in Spain, there were approximately 45,820 GPs in 2023, of whom around 20,000 are members of semFYC, and about 2,000 urologists, of whom 1,630 are members of the AEU. In addition, around 95% of clinicians in Spain work in the public sector, which is reflected in our sample. Nevertheless, the perspectives of non-affiliated or exclusively private-sector practitioners may be underrepresented. Future studies should aim to include a broader range of clinicians to enhance national representativeness. Moreover, our goal was not to achieve a proportional representation of all clinicians in Spain, but to allow a balanced comparison between general practitioners and urologists, which was the main objective of the study. Since primary care physicians outnumber urologists substantially in the Spanish health system, a proportionally representative sample would have resulted in a relatively small number of urologists, thereby limiting statistical power and reducing the ability to detect significant differences between specialties. For this reason, we set out to recruit sufficient numbers from both groups to ensure comparability in subgroup analyses. In addition, differential response rates between general practitioners and urologists may have influenced the results. However, sensitivity analyses-including specialty stratified analysis and multivariate models adjusted for key characteristics such as sex, years of practice, and prior PSA training-showed consistent results across models. The direction and statistical significance of major partnerships, particularly those related to shared decision-making and use of guidelines, remained stable, which suggests that the differential

response is unlikely to fully explain the observed differences based on speciality. Moreover, given the survey's national scope within Spain, caution must be exercised when generalizing findings internationally, considering variations in healthcare systems, clinical guidelines, and cultural factors affecting screening practices elsewhere. The response rate could not be determined, as the survey was distributed to all members of the participating scientific societies and responses were included consecutively until the required sample size was reached. This approach may introduce selection bias, as respondents could be more likely to represent a more motivated and engaged subgroup of clinicians, particularly those more interested in guideline-based practice or SDM. Consequently, the reported levels of SDM and other practices may be overestimated compared to routine clinical practice. Future studies using sampling strategies that allow calculation of response rates and broader inclusion of clinicians would improve representativeness. Nevertheless, this study includes a robust methodological framework, characterized by a well-defined sample size (n=494) and a comprehensive questionnaire development through an expert-validated modified Delphi method, enhancing the validity and reliability of the results. Additionally, the large and diverse sample of clinicians across different age groups and experience levels further strengthens its representativeness. Missing data ranged from 11% to 21% across variables, and although analyses suggested no significant differences between respondents with complete and incomplete data, the assumption that data were missing at random cannot be fully verified. However, sensitivity analyses using complete-case data showed consistent results, suggesting that missing data are unlikely to have substantially influenced the main findings.

## Conclusions

This study underscores significant, specialty-based variations in self-reported PSA screening practices between GPs and urologists, shaped by divergent perceptions, clinical responsibilities, and familiarity with guidelines. Addressing these disparities through targeted educational initiatives, simplified and harmonized clinical guidelines, and enhanced emphasis on SDM is essential to optimizing evidence-based PCa screening, ensuring patient-centered care, and ultimately improving healthcare outcomes.

## Supporting infomation

**S1 Table. STROBE Statement—Checklist of items that should be included in reports of cross-sectional studies.**
(DOC)

**S2 Questionnaire.**
(DOCX)

## Acknowledgments

We acknowledge the collaboration of the Spanish Society of Family and Community Medicine (semFYC) and the Spanish Association of Urology in the implementation of the survey, whose participation was essential for the development of the study.

## Author contributions

**Conceptualization:** Blanca Lumbreras, Lucy A. Parker, Pablo Alonso-Coello, Juan-Pablo Caballero-Romeu, Ignacio Párraga-Martínez, Luis Prieto, Irene Moral-Peláez, Mª del Campo-Giménez, Luis Gómez-Pérez, Ana Cebrián, Maite López-Garrigós, Elena Ronda, Mercedes Guilabert, Carlos Canelo-Aybar, Ildefonso Hernández-Aguado.

**Formal analysis:** Lucy A. Parker, Pablo Alonso-Coello, Juan-Pablo Caballero-Romeu.

**Methodology:** Blanca Lumbreras, Lucy A. Parker, Juan-Pablo Caballero-Romeu.

**Writing – original draft:** Blanca Lumbreras, Lucy A. Parker, Pablo Alonso-Coello, Juan-Pablo Caballero-Romeu.

**Writing – review & editing:** Ignacio Párraga-Martínez, Luis Prieto, Irene Moral-Peláez, Mª del Campo-Giménez, Luis Gómez-Pérez, Ana Cebrián, Maite López-Garrigós, Elena Ronda, Mercedes Guilabert, Carlos Canelo-Aybar, Ildefonso Hernández-Aguado.

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
