## [Decision Letter · Decision Letter 0]

29 Mar 2026

PONE-D-26-08019Disparities in prostate cancer screening practices among general practitioners and urologists (PROSHADE study): a cross-sectional studyPLOS One

Dear Dr. Lumbreras,

Thank you for submitting your manuscript to PLOS ONE. After careful consideration, we feel that it has merit but does not fully meet PLOS ONE’s publication criteria as it currently stands. Therefore, we invite you to submit a revised version of the manuscript that addresses the points raised during the review process.

We look forward to receiving your revised manuscript.

Kind regards,

Ismaheel Lawal, MD, PhD

Academic Editor

PLOS One

Journal Requirements:

“Research funded by the research project of the Instituto de Salud Carlos III, code PI20/01334, Principal Investigator Dr. Blanca Lumbreras Lacarra, co-financed with FEDER funds from the European Union "A way of doing Europe".”

3. In the online submission form, you indicated that “Dataset is available upon request to the authors.”

Reviewers' comments:

Reviewer's Responses to Questions

**Comments to the Author**

1. Is the manuscript technically sound, and do the data support the conclusions?

Reviewer #1: Partly

Reviewer #2: Partly

Reviewer #3: Partly

2. Has the statistical analysis been performed appropriately and rigorously? 

Reviewer #1: I Don't Know

Reviewer #2: No

Reviewer #3: Yes

3. Have the authors made all data underlying the findings in their manuscript fully available?

Reviewer #1: Yes

Reviewer #2: Yes

Reviewer #3: No

4. Is the manuscript presented in an intelligible fashion and written in standard English?

Reviewer #1: Yes

Reviewer #2: Yes

Reviewer #3: Yes

5. Review Comments to the Author

Reviewer #1: The manuscript addresses an important and timely topic, and the data presented are of clear interest. To further strengthen the paper and enhance its clarity and impact, several aspects could be refined.

The Introduction would benefit from an updated and more comprehensive European policy framework. In particular, it would be helpful to include reference to the EUCanScreen programme and to the Council Recommendation of 9 December 2022 on strengthening prevention through early detection (2022/C 473/01). This document explicitly highlights prostate cancer screening as an area where, in light of preliminary evidence and the widespread use of opportunistic screening, countries are encouraged to adopt a gradual approach, including pilot programmes and further research. It also specifies that PSA testing, combined with MRI as a follow-up test, should be evaluated in terms of feasibility and effectiveness within organised programmes. Framing the study within this context would reinforce its relevance, as it clearly aligns with the current European call for further scientific evidence in this field.

The section on the management of abnormal findings could be further clarified by considering how incidental findings, including those potentially outside the direct scope of urology, are addressed within the proposed pathway.

While the manuscript focuses on general practitioners and urologists, it might benefit from acknowledging the broader perspective of screening governance, particularly the role of public health specialists. These professionals contribute to ensuring adherence to shared protocols, quality assurance, and informed participation, which are key elements of organised screening programmes. Other specialists such as radiologists and pathologists play also a key role. In addition, the potential involvement of professional associations and patient organisations in the co-construction of screening strategies could be briefly mentioned as a possible future development. Their inclusion in a further Delphi process, for example, might offer valuable complementary perspectives.

Some sentences in the final part of the Introduction appear slightly redundant and could be streamlined to improve readability.

The Ethics section would also benefit from clarification. As the study does not involve human subjects, identifiable data, or interventions, it may be sufficient to state that compliance with the Declaration of Helsinki was not required. In addition, further detail on the process of obtaining written informed consent for participation in the Delphi survey would improve transparency, particularly considering that the questionnaire was distributed via mailing lists and completed through an anonymous Google Form. If available, providing the consent form as supplementary material could be helpful.

From a methodological perspective, it may be useful to emphasise more clearly that the study captures self-reported practices and intentions, rather than observed behaviours. Adopting more cautious wording throughout the manuscript, including the Discussion, could help maintain consistency and avoid overinterpretation. This is particularly relevant when comparing general practitioners and urologists, who operate in different clinical contexts and interact with different patient populations. For example, the reported consistency of annual PSA testing might reflect an intention to prescribe rather than actual practice, especially among urologists working predominantly in public settings (95% of the sample), where patients are more likely to present already with (urologic) symptoms or specific clinical indications. In contrast, general practitioners are more likely to engage with the general population and preventive care, making their reported prescribing intentions more closely aligned with real-world screening practices. This distinction could be more explicitly considered when interpreting the findings.

Some results would benefit from a more nuanced interpretation. In particular, clinicians who reported receiving formal education on PSA testing appear less likely to consider the test useful for diagnosis and more likely to communicate its limitations to patients. This is a relevant finding that could be more clearly highlighted and consistently reflected in the Discussion. In this light, conclusions suggesting that educational programmes should primarily aim to increase PSA screening uptake among general practitioners may not be fully supported by the data. Rather, the findings seem to suggest that education is associated with a more cautious and balanced approach, including greater attention to the potential harms of screening.

Related to this, the statement that knowledge of clinical guidelines improves the completeness and balance of information provided to patients could be reconsidered in light of the results, which seem to indicate that clinicians more familiar with guidelines may still be less likely to discuss key aspects such as overdiagnosis, overtreatment, and the use of shared decision-making, aspects that have already contributed in the past to slow the initial enthusiasm in prostate cancer screening. Given that shared decision-making is widely recognised as essential in contexts where recommendations are not uniform, this aspect could be further explored.

In discussing variability in guidelines, it might also be useful to briefly acknowledge that different types of guideline-producing bodies may adopt different methodological approaches. For example, public health-oriented organisations (USPSTF) aim to minimise potential conflicts of interest by limiting the direct involvement of specialists in decision-making panels, whereas specialty-driven guidelines may reflect a more focused clinical perspective.

The results concerning clinicians who received formal education on PSA testing and their lower concern about underdiagnosis could be clarified, as the current phrasing may suggest a greater inclination towards screening, whereas the data appear to indicate a more cautious stance. This point is particularly important and could support a more consistent and balanced discussion. In this context, it may also be worth considering that, while general practitioners appear to integrate communication of risks more frequently, urologists, who often work in more specialised settings, might benefit from additional support and tools for effective communication with patients, particularly regarding overdiagnosis and overtreatment.

Some minor issues of consistency and clarity should also be addressed, including formatting (e.g., lines 316, 419, 432), consistent definition and use of acronyms (e.g., USPSTF, PAPSP), and uniform terminology for figures and graphs (e.g., Figure vs Graph, Figure 1A/1B).

Overall, the study provides interesting insights into this complex and evolving topic but I would suggest a more cautious and closely aligned interpretation of the findings, together with improved contextualisation and consistency to help the manuscript better reflect its strengths and enhance its contribution to the discussion.

Reviewer #2: Thank you fir submitting your work. Please find my comments:

1. Recruitment only from societies likely introduces 'volunteer bias'. Private sector / non-affiliated / remote area practitioners may be less engaged with guideline-based practices. This leads to limited national generalizability. Please address this limitation, if cannot be performed.

2. Response rate is required among the physicians invited to participate. If its too low, the cohort may reflect motivated minority, rather than typical practice. Real-world SDM % may be lower than reported. Please address this.

3. Statistical confounding: GPs and Urologists are not comparable groups, which differ not just in knowledge, but also in practice setting and patient-approach. Chi-square would likely cause unadjusted comparison. Logistic regression is recommended for SDM, PSA screening and guidelines-adherence for multiple predictors (such as specialty, sex, years of experience etc.). Is specialty significant after adjustment? This would be a true measure of specialty-based practice differences.

4. Expand on limitation of cross-sectional study: Knowledge of guidelines lead to behavior change cannot be inferred, only association can be commented upon. Please modify verbiage accordingly.

5. More knowledge but less SDM for Urologist: A powerful but underdeveloped finding of this study. Authors should give their interpretation on why this paradox exists, pertaining to their local community / culture. Such intellectual interpretation could help others to expand on public health research and education.

5. Minor grammatical corrections: GPS to GPs, UAE to EAU in table 3.

Reviewer #3: This manuscript reports findings from the PROSHADE cross-sectional survey, which examines the knowledge, attitudes, and practices (KAP) of general practitioners and urologists in Spain regarding PSA testing and prostate cancer screening. The topic is clinically relevant and timely. However, the manuscript presents several methodological limitations, reporting inconsistencies, and analytical issues that need to be addressed through major revisions before it can be considered for publication in PLOS One.

Major concerns:

It is mentioned that there are 1,630 urologists included in the AEU and 20,000 GPs in the SEMFYC. There is a large difference in sampling fractions between the two groups - 214 out of the 1,630 urologists versus 280 out of the 20,000 GPs took the survey. Response rate is not reported anywhere in the manuscript, which is a critical omission for survey research. The discrepancy in sampling fractions between urologists and GPs (13% vs. 1.4%) raises concerns about selection bias and representativeness. The authors should report response rates explicitly and discuss potential non-response bias. Consider sensitivity analyses to address differential response rates.

The authors consecutively selected the first clinicians who answered the survey. This describes a convenience sample, not a probability sample. However, the sample size calculation (lines 220-229) assumes simple random sampling to achieve 5% precision with 95% CI. This mismatch undermines the validity of the precision estimates. The authors should acknowledge that this is a convenience sample and revise claims about precision accordingly.

Tables 1, 2 and 3 demonstrate multiple chi-square tests without any correction for multiple testing. This increases the risk of Type I error. Consider applying corrections such as Bonferroni correction.

The authors mention that chi-square tests were used for all associations. However, Table 1 includes continuous variables like age and years of practice with median comparisons. Chi-square is inappropriate for continuous data. Please clarify which tests were used for continuous variables and report test statistics appropriately. For continuous data, tests of normality should be performed before choosing the appropriate test for comparison.

The analyses presented are all bivariate, comparing GPs and urologists without adjusting for other factors. Observed differences may be influenced by potential confounders such as age, sex, years of practice, or prior training. For example, if urologists are younger or more recently trained, differences may reflect generational or educational factors rather than specialty per se. The manuscript would be strengthened if the authors could conduct multivariable regression analyses that adjust for these potential confounders to assess the independent association of specialty with the outcomes of interest.

The authors assume data are missing at random without providing evidence. Missing data ranges from 11-21% across questions. No sensitivity analyses were performed. Please justify the MAR assumption with supporting analyses or conduct sensitivity analyses.

All answers were presented as Likert scale. However, Tables 2-3 show categorical/nominal response options as well. Please clarify in methods section.

Table 1 collapses Likert responses into binary categories: "none or little" versus "quite a lot or a lot" without methodological justification. This dichotomization discards information and reduces statistical power. Please justify the dichotomization approach or present ordinal analyses.

Line 72: Can a descriptive cross-sectional survey identify "associations"? This language suggests analytical epidemiology, but the design is primarily descriptive. Consider revising to "examine differences" or "compare."

Lines 366-367 states that "knowledge of clinical guidelines positively influences the completeness and balance of information" which could be interpreted as implying a causal relationship. Given the cross-sectional design of the study, causality cannot be established. It would be helpful for the authors to rephrase this statement using language that more accurately reflects an association rather than a causal effect.

Lines 421-433 covering the limitations section is brief and does not address the major methodological concerns outlined above. The authors should expand the limitations section to address the issues described above.

Moderate concern:

Although this is a KAP study, the authors did not use the ChecKAP reporting checklist, which was specifically developed for this study type and includes 46 items across 8 domains. The authors should complete and submit the ChecKAP checklist as supplementary material to ensure comprehensive reporting of all essential KAP study elements.

Minor concerns:

Formatting/typographical errors:

Line 152: Reference appears in different font/size.

Line 316: Font/size formatting inconsistency.

Lines 432-433: Font/size formatting inconsistency.

Line 337 - "uptake" should likely be "update" when referring to guideline updates.

In Table 3 - "UAE" should be "EAU" (European Association of Urology)

Line 458: Missing close bracket after SemFyc.

All tables: Use footnotes to explain abbreviations such as NA, NS/NC

Line 187: Use "cross-sectional survey" instead of "cross-sectional study."

Table 1: Clarify what "regular formation" means.

6. PLOS authors have the option to publish the peer review history of their article (what does this mean?). If published, this will include your full peer review and any attached files.

Reviewer #1: No

Reviewer #2: No

Reviewer #3: **Yes:** Jemimah Nayar

---

## [Author Response · Author response to Decision Letter 1]

2 May 2026

PONE-D-26-08019

Disparities in prostate cancer screening practices among general practitioners and urologists (PROSHADE study): a cross-sectional study

Thank you very much for your comments and interest in the manuscript.

Journal requirements

- According to the journal requirements, we have included hte following sentence: "The funders had no role in study design, data collection and analysis, decision to publish, or preparation of the manuscript." (page 24, lines 451-453).

- In the online submission form, you indicated that “Dataset is available upon request to the authors.”. We have included the following sentence: “All relevant data are within the manuscript and its Supporting Information files. The dataset generated and analyzed during the current study is available in Zenodo at https://doi.org/10.5281/zenodo.19983225”.

Reviewer #1: The manuscript addresses an important and timely topic, and the data presented are of clear interest. To further strengthen the paper and enhance its clarity and impact, several aspects could be refined.

The Introduction would benefit from an updated and more comprehensive European policy framework. In particular, it would be helpful to include reference to the EUCanScreen programme and to the Council Recommendation of 9 December 2022 on strengthening prevention through early detection (2022/C 473/01). This document explicitly highlights prostate cancer screening as an area where, in light of preliminary evidence and the widespread use of opportunistic screening, countries are encouraged to adopt a gradual approach, including pilot programmes and further research. It also specifies that PSA testing, combined with MRI as a follow-up test, should be evaluated in terms of feasibility and effectiveness within organised programmes. Framing the study within this context would reinforce its relevance, as it clearly aligns with the current European call for further scientific evidence in this field.

We thank the reviewer for this suggestion. In line with this comment, we have revised the Introduction to incorporate an updated European policy framework. Specifically, we have included references to the EUCanScreen programme and the Council Recommendation of 9 December 2022 on strengthening prevention through early detection (introduction section, pages 5-6, lines 131-148).

The section on the management of abnormal findings could be further clarified by considering how incidental findings, including those potentially outside the direct scope of urology, are addressed within the proposed pathway.

We thank the reviewer for this comment. In response, we have expanded the section on the management of abnormal findings to address incidental findings, including those that may fall outside the direct scope of urology (introduction section, page 6, lines 159-162).

While the manuscript focuses on general practitioners and urologists, it might benefit from acknowledging the broader perspective of screening governance, particularly the role of public health specialists. These professionals contribute to ensuring adherence to shared protocols, quality assurance, and informed participation, which are key elements of organised screening programmes. Other specialists such as radiologists and pathologists play also a key role. In addition, the potential involvement of professional associations and patient organisations in the co-construction of screening strategies could be briefly mentioned as a possible future development. Their inclusion in a further Delphi process, for example, might offer valuable complementary perspectives.

We thank the reviewer for this suggestion. In response, we have included a paragraph in the manuscript to acknowledge the broader governance perspective of organised screening programmes beyond the roles of general practitioners and urologists. (introduction section, page 7, lines 176-185).

Some sentences in the final part of the Introduction appear slightly redundant and could be streamlined to improve readability.

We thank the reviewer for this observation. In response, we have revised the final part of the Introduction to remove redundant sentences and improve clarity and readability.

The Ethics section would also benefit from clarification. As the study does not involve human subjects, identifiable data, or interventions, it may be sufficient to state that compliance with the Declaration of Helsinki was not required. In addition, further detail on the process of obtaining written informed consent for participation in the Delphi survey would improve transparency, particularly considering that the questionnaire was distributed via mailing lists and completed through an anonymous Google Form. If available, providing the consent form as supplementary material could be helpful.

We thank the reviewer for this suggestion. In response, we have revised the Ethics section to clarify that the study does not involve human subjects, identifiable personal data, or clinical interventions, and therefore formal compliance with the Declaration of Helsinki was not required. We have also expanded the description of the informed consent process to improve transparency. The revised text now specifies that participants were invited via professional mailing lists, received detailed information about the study, and that completion of the anonymous online questionnaire (administered through Google Forms) was considered as provision of informed consent. (Material and method section, page 13, lines 346-354).

From a methodological perspective, it may be useful to emphasise more clearly that the study captures self-reported practices and intentions, rather than observed behaviours. Adopting more cautious wording throughout the manuscript, including the Discussion, could help maintain consistency and avoid overinterpretation. This is particularly relevant when comparing general practitioners and urologists, who operate in different clinical contexts and interact with different patient populations. For example, the reported consistency of annual PSA testing might reflect an intention to prescribe rather than actual practice, especially among urologists working predominantly in public settings (95% of the sample), where patients are more likely to present already with (urologic) symptoms or specific clinical indications. In contrast, general practitioners are more likely to engage with the general population and preventive care, making their reported prescribing intentions more closely aligned with real-world screening practices. This distinction could be more explicitly considered when interpreting the findings.

Thank you to the reviewer for this comment. In response, we have reviewed the manuscript to emphasize more clearly that findings reflect self-declared practices and intentions rather than directly observed clinical behaviors. In addition, we have expanded the Discussion to better contextualize the comparison between general practitioners and urologists. The revised text highlights differences in their clinical settings and patient populations, noting that urologists (predominantly working in public and specialized care) are more likely to see patients with specific symptoms or indications, while general practitioners are more committed to preventive care and screening at the population level. This distinction is now explicitly taken into account when interpreting the findings, particularly with respect to PSA test patterns.

The corresponding changes have been incorporated throughout the manuscript:

- Page 3: lines 78; 79; 80; 82.

- Page 4: line 91.

- Page 16: lines 407; 417.

- Page 17, lines 431-433; 437-440; 441; 444; 445; 448-451.

- Page 21: lines 485; 486; 491; 493; 498; 500; 502.

- Page 25: lines 558; 564; 565.

- Page 26: lines 589; 608.

- Page 27: lines 623; 624; 641; 642; 643

- Page 28: lines 660; 66; 665.

- Page 31: line 741.

Some results would benefit from a more nuanced interpretation. In particular, clinicians who reported receiving formal education on PSA testing appear less likely to consider the test useful for diagnosis and more likely to communicate its limitations to patients. This is a relevant finding that could be more clearly highlighted and consistently reflected in the Discussion. In this light, conclusions suggesting that educational programmes should primarily aim to increase PSA screening uptake among general practitioners may not be fully supported by the data. Rather, the findings seem to suggest that education is associated with a more cautious and balanced approach, including greater attention to the potential harms of screening.

Thank you for this comment. We agree that this is an important aspect of our findings that deserves clearer emphasis in the Discussion. In line with your suggestion, we have revised the manuscript to better highlight that clinicians who reported receiving formal education on PSA testing were less likely to consider the test useful for diagnostic purposes, more likely to communicate its limitations to patients, and less concerned about under-diagnosis of prostate cancer. (Discussion section, page 26, lines 596-606).

Related to this, the statement that knowledge of clinical guidelines improves the completeness and balance of information provided to patients could be reconsidered in light of the results, which seem to indicate that clinicians more familiar with guidelines may still be less likely to discuss key aspects such as overdiagnosis, overtreatment, and the use of shared decision-making, aspects that have already contributed in the past to slow the initial enthusiasm in prostate cancer screening. Given that shared decision-making is widely recognised as essential in contexts where recommendations are not uniform, this aspect could be further explored.

Thank you for your comment. We have included a detailed discusión of this point comparing urologists and GPs (Discussion section pages 25-26, lines 571-588).

In discussing variability in guidelines, it might also be useful to briefly acknowledge that different types of guideline-producing bodies may adopt different methodological approaches. For example, public health-oriented organisations (USPSTF) aim to minimise potential conflicts of interest by limiting the direct involvement of specialists in decision-making panels, whereas specialty-driven guidelines may reflect a more focused clinical perspective.

Thank you for this suggestion. We agree that acknowledging differences in the methodological approaches of guideline-producing bodies adds important context to the observed variability in clinical practices. In response, we have revised the paragraph to include a sentence noting that public health-oriented organisations, such as the USPSTF, often aim to minimise potential conflicts of interest by limiting specialist involvement in decision-making panels, whereas specialty-driven guidelines may reflect a more focused clinical perspective. (Discussion section page 27, lines 632-637).

The results concerning clinicians who received formal education on PSA testing and their lower concern about underdiagnosis could be clarified, as the current phrasing may suggest a greater inclination towards screening, whereas the data appear to indicate a more cautious stance. This point is particularly important and could support a more consistent and balanced discussion. In this context, it may also be worth considering that, while general practitioners appear to integrate communication of risks more frequently, urologists, who often work in more specialised settings, might benefit from additional support and tools for effective communication with patients, particularly regarding overdiagnosis and overtreatment.

Thank you for the comment. We have revised the section to clarify that these clinicians were, in fact, less concerned about underdiagnosis while simultaneously demonstrating a more cautious and balanced perspective, including greater awareness of the limitations and potential harms of screening. Taking into account this comment and the previous one, this interpretation is now more aligned with the data and presents a more consistent perspective. (Discussion section, page 26, lines 596-606).

Some minor issues of consistency and clarity should also be addressed, including formatting (e.g., lines 316, 419, 432), consistent definition and use of acronyms (e.g., USPSTF, PAPSP), and uniform terminology for figures and graphs (e.g., Figure vs Graph, Figure 1A/1B).

Done.

Overall, the study provides interesting insights into this complex and evolving topic but I would suggest a more cautious and closely aligned interpretation of the findings, together with improved contextualisation and consistency to help the manuscript better reflect its strengths and enhance its contribution to the discussion.

Thank you for this overall assessment. We agree with the need for a more cautious and closely aligned interpretation of the findings. In response, we have revised the manuscript to ensure that the conclusions more accurately reflect the data, avoiding overinterpretation and better emphasising the nuances observed.

Reviewer #2: Thank you fir submitting your work. Please find my comments:

1. Recruitment only from societies likely introduces 'volunteer bias'. Private sector / non-affiliated / remote area practitioners may be less engaged with guideline-based practices. This leads to limited national generalizability. Please address this limitation, if cannot be performed.

Thank you for this comment. We acknowledge that recruiting participants exclusively through scientific societies such as the Spanish Association of Urology (AEU) and the Spanish Society of Family and Community Medicine (semFYC) may introduce a degree of volunteer bias. Clinicians affiliated with these organisations could be more likely to be more engaged with continuing education, guideline updates, and professional development activities than non-affiliated practitioners. At the same time, it is important to contextualise our sampling approach. In Spain, there were approximately 45,820 GPs in 2023, of whom around 20,000 are members of semFYC. Similarly, there are approximately 2,000 practising urologists in Spain, of whom 1,630 are members of the AEU. Therefore, although not fully representative, recruitment through these societies allowed access to a large and geographically diverse pool of clinicians. The aim of this approach was to facilitate participation from different regions across Spain. In addition, approximately 95% of clinicians in Spain worked in the public sector. We have addressed this limitation in the revised manuscript and highlighted the need for future studies to include a broader and more diverse sample of practitioners to enhance national representativeness. (Discussion section, page 29, lines 690-700).

2. Response rate is required among the physicians invited to participate. If its too low, the cohort may reflect motivated minority, rather than typical practice. Real-world SDM % may be lower than reported. Please address this.

Thank you for this comment. We agree that the response rate is a relevant aspect when interpreting the representativeness of the findings. In our study, the survey was distributed to all members of the participating scientific societies (AEU and semFYC), and we included consecutively the first responses received until the required sample size was reached. Based on our sample size calculation, a minimum of 384 participants was needed to ensure adequate precision, and we ultimately included 494 clinicians. However, we acknowledge that this approach does not allow for the calculation of a precise response rate and may introduce selection bias. It is possible that clinicians who chose to respond were more motivated, more engaged with guideline-based practices, or more interested in shared decision-making (SDM), which could lead to an overestimation of its implementation in routine practice. As such, the reported levels of SDM may not fully reflect real-world practice among all clinicians. We have addressed this limitation in the revised manuscript by explicitly acknowledging the potential for selection bias. Future studies using sampling strategies that allow for response rate estimation and broader inclusion of clinici

---

## [Editor Report · Decision Letter 1]

5 May 2026

Disparities in prostate cancer screening practices among general practitioners and urologists (PROSHADE study): a cross-sectional study

PONE-D-26-08019R1

Dear Dr. Lumbreras,

We’re pleased to inform you that your manuscript has been judged scientifically suitable for publication and will be formally accepted for publication once it meets all outstanding technical requirements.

Kind regards,

Ismaheel Lawal, MD, PhD

Academic Editor

PLOS One
---

## [Editor Report · Acceptance letter]

PONE-D-26-08019R1

PLOS One

Dear Dr. Lumbreras,

I'm pleased to inform you that your manuscript has been deemed suitable for publication in PLOS One. Congratulations! Your manuscript is now being handed over to our production team.

Kind regards,

on behalf of

Dr. Ismaheel Lawal

Academic Editor

PLOS One